# The potential of sea ice leads as a predictor for summer Arctic sea ice extent

Yuanyuan Zhang[1,] Xiao Cheng[1], Jiping Liu[2], Fengming Hui[1]

[1]State Key Laboratory of Remote Sensing Science, and College of Global Change and Earth System Science, and Joint Center for Global Change Studies, Beijing Normal University, Beijing 100875, China
[2]Department of Atmospheric and Environmental Sciences, University at Albany, State University of New York, Albany, NY 12222, USA

*Correspondence to*: Xiao Cheng (xcheng@bnu.edu.cn)

**Abstract.** The Arctic sea ice extent throughout the melt season is closely associated with initial sea ice state in winter and spring. Sea ice leads are important sites of energy fluxes in the Arctic Ocean, which may play an important role in the evolution of Arctic sea ice. In this study, we examine the potential of sea ice leads as a predictor for summer Arctic sea ice extent forecast using a recently developed daily sea ice leads product retrieved from Moderate-Resolution Imaging Spectroradiometer. Our results show that July pan-Arctic sea ice extent can be predicted from the area of sea ice leads integrated from mid-winter to late spring with the prediction error of 0.28 million km$^2$ that is smaller than the standard deviation of the observed interannual variability. However, the predictive skills for August and September pan-Arctic sea ice extent are very low. When the area of sea ice leads integrated in the Atlantic and central and west Siberian sector of the Arctic is used, it has a significantly strong relationship (high predictability) with both July and August sea ice extent in the Atlantic and central and west Siberian sector of the Arctic. Thus, the realistic representation of sea ice leads (e.g., the areal coverage) in numerical prediction systems might improve the skill of forecast in the Arctic region.

## 1 Introduction

Sea ice is an important component of the climate system. In the past few decades, Arctic sea ice has undergone dramatic change associated with changes in atmospheric and oceanic processes (Comiso et al., 2008; Ding et al., 2017; Liu et al., 2013; Parkinson and Comiso, 2013; Richter-Menge et al., 2016; Stroeve et al., 2007; Stroeve et al., 2012). Satellite observation shows a decreasing Arctic sea ice extent at an annual rate of about 4.73% per decade and a faster rate of 13.56% per decade in September for the period of 1979-2017, calculated using the Arctic sea ice index obtained from the National Snow and Ice Data Center (see data section for details). The decreasing Arctic sea ice not only affects the local environment and community, i.e., brings opportunities and challenges to indigenous people (Forbes et al., 2016; Lamers et al., 2016), but also has strong feedback on other components of the climate system, i.e., increases the frequency of abnormal weather and climate in the mid-latitude of the northern hemisphere and influences the thermohaline circulation (Budikova, 2009;

Levermann et al., 2007; Liu et al., 2012; Vihma, 2014). Hence there is an increasing demand for Arctic sea ice prediction at seasonal and longer timescales, especially during the melting season (Eicken, 2013; Stroeve et al., 2014).

Many works have been done to improve seasonal forecast of Arctic sea ice (Guemas et al., 2016; Stroeve et al., 2014). Seasonal prediction of Arctic sea ice extent has been produced with statistical methods, i.e., many use regression type statistical models, trained from historical data and then applied to forecast the near future. To date, statistical models show comparable or slightly higher skill than dynamic models in terms of the prediction of the total Arctic sea ice extent (Blanchard-Wrigglesworth et al., 2015; Stroeve et al., 2014). The evolution of Arctic sea ice extent during summer and fall is closely associated with initial sea ice conditions in winter and spring. The potential of different sea ice parameters as predictors of Arctic sea ice extent has been explored using empirical statistical models. The results show that some parameters can significantly contribute to the improvement in seasonal sea ice forecast at different lead times (Holland and Stroeve, 2011; Lindsay et al., 2008). For example, sea ice concentration and surface temperature in spring are introduced into a multiple linear regression model to forecast the minimum Arctic sea ice extent (Drobot, 2007; Drobot et al., 2006). Some studies suggested that accurate sea ice thickness can increase forecast skill 2-month ahead (Day et al., 2014; Dirkson et al., 2017). Recently, the spring melt pond fraction has been employed to improve the skill of forecasting September sea ice extent (Liu et al., 2015; Schröder et al., 2014). An annual sea ice outlook has solicited prediction of mean September Arctic sea ice extent from the research community since 2008. The result shows that the median sea ice predictions are off by a large margin in 2009, 2012 (record low), 2013, 2014 and 2016 (second record low) (Hamilton and Stroeve, 2016; Stroeve et al., 2014).

Sea ice leads develop as quasi-rectilinear cracks within the ice pack due to sea ice dynamics and warm water upwelling at particular locations. Leads can be kilometres to tens of kilometres long and meters to kilometres wide, which are more prevalent in areas of thin ice (i.e., the marginal ice zone) than in the central Arctic ice pack (Wadhams et al., 1985). Though leads only cover a small proportion of sea ice area, they are important sites of energy fluxes that can cause a large fluctuation of air temperature (Lüpkes et al., 2008). Leads are responsible for about 50% of a transfer of sensible heat from the Arctic Ocean to the atmosphere during winter (Maykut, 1982). In-situ measurements from the Arctic Ice Dynamics Joint Experiment Sea Ice Lead Experiment in 1974 showed that sensible heat and latent fluxes from leads can exceed 400 W m$^{-2}$ and 130 W m$^{-2}$ respectively (Andreas et al., 1979). Sensible heat flux over sea ice leads depends strongly on leads' width. Narrow leads are over two times more efficient in transferring heat than larger ones (Maykut, 1982). When considering the leads' width influence into assessment, heat flux can be up to 55% larger (Marcq and Weiss, 2011). In additional, the albedo of leads is about 0.07 under cloudy condition (Tschudi et al., 2002) in contrast to 0.6-0.9 of sea ice or snow-covered ice (Perovich et al., 2002). As a result, the leads absorb more shortwave radiation. Adversely, sea ice leads that persist throughout the winter are often accompanied by low-level clouds downwind because of the release of heat and moisture into the atmosphere, influencing surface energy budget.

While sea ice leads play an important role in the determination of the evolution of Arctic sea ice, their potential role in Arctic sea ice prediction have not been examined. One reason is that lack of observations sea ice leads with sufficient spatial and temporal coverage. This hampers our understanding of variability of sea ice leads in the Arctic Ocean, and their relationship with Arctic sea ice cover (Ivanova et al., 2016; Wernecke and Kaleschke, 2015). Another reason is that sea ice leads are unrepresented process in numerical prediction systems and climate models due to their highly nonlinear, small-scale, and intermittent characteristics (Spreen et al., 2017; Wang et al., 2016). As a result, potential effects of sea ice leads on Arctic sea ice prediction are not well understood. Given that sea ice leads dominate the atmosphere-sea ice-ocean interface in the aforementioned manner, in this study, we use a recently developed sea ice leads product retrieved from the Moderate-Resolution Imaging Spectroradiometer thermal infrared data to examine the potential of sea ice leads as a predictor for summer Arctic sea ice extent forecast.

## 2 Data and Methods

Compared to abundant research on characterizing variability of Arctic sea ice concentrations (Cavalieri and Parkinson, 2012; Comiso et al., 2008), there are limited efforts focused on characterizing the variability of sea ice leads in the Arctic Ocean. Some progress has been made on the detection of sea ice leads using remote sensing imagery. Lindsay and Rothrock (1995) conducted a semi-automatic detection of sea ice leads, in which the Advanced Very High Resolution Radiometer (AVHRR) images are used to distinguish the potential of open water/leads through spectral unmixing analysis, and the uncertainty is mainly depended on manual cloud remove procedure. Miles and Barry (1998) manually mapped a five-year sea ice leads climatology for the Arctic Ocean using the Defense Meteorological Satellite Program (DMSP) thermal and visible band imagery. SAR or microwave imagery can be used to obtain sea ice surface details with minimum cloud influence. Kwok (1998) used the RADARSAT Geophysical Processor System (RGPS) to estimate the deformation of sea ice and identify the linear kinematics features - sea ice leads. Röhrs and Kaleschke (2010) presented an algorithm applied to the passive microwave imagery from the Advanced Microwave Scanning Radiometer for EOS (AMSR-E) to detect sea ice leads wider than 3 km. The RGPS and AMSR-E sea ice lead products have been validated in the entire Arctic Ocean and have the capability to show spatial variability of sea ice leads (Bröhan and Kaleschke, 2014; Kwok and Cunningham, 2002).

In a recent study, Willmes and Heinemann (2015a) presented a non-parameterized global threshold method, which was validated and applied to derive sea ice leads maps from surface temperature anomalies in the Arctic Ocean using the MODIS ice surface temperature product. Daily sea ice leads composites were created. The composite maps indicate the presence of cloud artifacts in the leads identification that arise from ambiguities in the MODIS cloud mask. To mitigate these artifacts, they implemented a fuzzy filter system that employs spatial and temporal object characteristics to distinguish between physical leads and artifacts. This approach advances the potential to retrieve daily leads maps operationally from the MODIS infrared product.

In this study, the pan-Arctic sea ice leads data is obtained from the Data Publisher for Earth & Environment Science (PANGAEA), which is available for the months from January to April for the period 2003-2015 (Willmes and Heinemann, 2015b). The spatial resolution of the daily binary sea ice leads map is about 1.5 km with omission 5% that can reflect sea ice leads variability except the Chukchi Sea (Willmes and Heinemann, 2015a, c), because clear-sky day is less than 15% in the

Chukchi Sea. Cloud contamination is a major issue plaguing the retrieval of the pan-Arctic sea ice leads from the MODIS infrared observation. Here we compare the above MODIS sea ice leads data with the Synthetic Aperture Radar (SAR) images under cloudy conditions. Compared to MODIS that receives thermal emissions or reflected components, SAR allows for penetration through most clouds and precipitation. We calculate backscatter coefficients from the Sentinel-1A Extra-Wide swath HH polarization images using the Sentinel Application Platform and project them on the NSIDC polar-

stereographic grid with a spatial resolution of 100 m. Cloudy conditions are determined using the MOD08 Level3 daily cloud fraction product (Hubanks et al., 2018). For example, Figure 1 shows the MODIS cloud fraction, SAR backscatter coefficient image, and MODIS sea ice leads in the northern Beaufort Sea on April 11, 2015. Compared to SAR images, the MODIS sea ice leads data can capture the correct spatial distribution of sea ice leads under cloudy conditions. The consistence between the MODIS sea ice leads data and SAR image gives us more confidence about this data.

The Arctic sea ice extent is obtained from the National Snow and Ice Data Center (NSIDC), which is derived from the Nimbus-7 Scanning Multichannel Microwave Radiometer, DMSP Special Sensor Microwave/Imager, and Special Sensor Microwave Imager and Sounder sensors using NASA Team algorithm (Cavalieri et al., 1996, updated yearly).

The daily total area of sea ice leads is computed from the daily binary sea ice leads map, which is projected on the NSIDC polar-stereographic grid with a spatial resolution of 25 km. During the projection, we calculate the number of pixels

with detected sea ice leads in a 25 km grid box. Sea ice leads fraction is then defined as the ratio between the number of pixels with detected sea ice leads and the total number of pixels in the 25 km grid box. The total area of sea ice leads is the sum of the product of the sea ice leads fraction and the area of the grid box ($625 \text{ km}^2$). Here the daily total area of sea ice leads is only calculated when the NSIDC sea ice concentration in the grid box is larger than 15% (commonly used as the threshold to define sea ice edge).

**3 Results**

Figure 2 shows the evolution of the daily total area of sea ice leads in the Arctic Ocean from January 1 to April 30 averaged for the period of 2003-2015. As Superimposed on large year-to-year variation for each single day as shown by the grey shaded in Figure 2, the climatology of the total sea ice lead area exhibits a gradually decrease from ~0.8 million $\text{km}^2$ in early January to ~0.5 million $\text{km}^2$ in late April. As shown in Figure 3a, overall, there is no significant trend in the total area of sea

ice leads averaged for January-April during 2003-2015, although it shows an increasing tendency from 2003 to 2013. The year 2013 had the largest area of sea ice leads (0.91 million $\text{km}^2$) followed by the smallest area in the year 2014 (0.45 million

km$^2$). We also calculate the correlation coefficients between July, August, September sea ice extent and the area of sea ice leads averaged from January to April during 2003-2015, which are -0.51, -0.30 and -0.23, respectively. It appears that July sea ice extent is more closely related to the area of sea ice leads than August and September. Figure 3b shows the spatial distribution of the trend of the sea ice lead area in each individual 25 km grid box. The area of sea ice leads has exhibited an

increasing trend extending from the Greenland Sea, through the northern Barents Sea, to the Laptev and Kara Seas, and a decreasing trend in the southern Barents Sea, between the eastern Siberian Sea and Chukchi Sea, and along the coast of Alaska. In particular, the strong out of phase trend between the northern and southern Barents Sea is persistent for each individual month. However, most of these trends are not significant at 95% confidence level, except the southern Barents Sea.

To investigate the relationship between the area of sea ice leads in the Arctic Ocean from late winter to mid-spring and Arctic sea ice extent during the melting season, we calculate the correlation between the time series of the sea ice lead area averaged for January-April and sea ice extent in July, August and September, respectively, during 2003-2015. It should be noted that when examining correlation between two variables with large trends. Two variables might be linked statistically but physically independent. Thus, we remove the trend for all time series before calculating the correlation.

Following similar procedures in Liu et al. (2015), we integrate the area of sea ice leads in the Arctic Ocean over time and space to generate the sea ice lead time series. Specifically, first, we integrate the average area of sea ice leads occurring in each individual 25 km grid point varying from 1 January to 2 January, to 3 January, and up through 30 April. Second, we calculate the correlation coefficient between the de-trended time series of the integrated area of sea ice leads at each grid point and the de-trended time series of the total sea ice extent in July, August and September, respectively. As

discussed earlier, in general, more sea ice leads during late winter to mid-spring, even when they refreeze, tend to contain thinner and weaker sea ice that are more susceptible to atmospheric winds (i.e., storm) and air temperature (i.e., warm advection). This may result in less sea ice during the melting season. Thus, more sea ice leads are expected to negatively associate with following sea ice extent, so we expect negative correlations between sea ice leads and sea ice extent.

Figure 4 shows spatial correlation maps between the area of sea ice leads integrated from 1 January to the day given

and the total Arctic sea ice extent in July, August and September, respectively. For July sea ice extent (Fig. 4 left column), some small clusters of significant negative correlations but scattered are found in the Arctic Ocean north of ~75°N as the area of sea ice leads is integrated for one month from 1 January to 30 January (black crosses in Fig. 4a). These small clusters become relatively broader as the area of sea ice leads is integrated to the end of February (60 days, Fig. 4b), covering a relatively larger percentage of the central Arctic Ocean as well as much of the western Greenland Sea and northern Barents

and Kara Seas. By the end of March, extending the integration to 90 days, the area with significant correlations is enlarged remarkably, especially in the Atlantic and central and west Siberian sector of the Arctic (Fig. 4c). Extending the integration to the end of April (120 days), the area with significant correlations has minimal change (Fig. 4d) compared to that of Figure 4c. The spatial distribution of significant correlations for August and September sea ice extent is similar (Fig. 4, 2$^{nd}$ and 3$^{rd}$

column).A small cluster of significant negative correlations is found in the western Laptev Sea as the area of sea ice leads is integrated for one month (Fig. 4e and Fig. 4i). The cluster becomes broader and extends northward into central Arctic Ocean after the 2-month integration (Fig. 4f and Fig. 4j). Extending the integration time period beyond March yields only small change in the area with significant negative correlations (Fig. 4h and Fig. 4l).

5        Here we generate time series of the total area of sea ice leads integrated from 1 January to 2 January, to 3 January, up to 30 April for the grid points having significant negative correlation coefficients between sea ice leads integrated from 1 January to 30 April and July Arctic sea ice extent (grid points with black crosses in Fig. 4d). We then calculate the correlation between time series integrated to the day given and time series of July Arctic sea ice extent. As shown in Figure 5 (blue line), the correlation between sea ice leads and July sea ice extent is not statistically significant at the 99% confidence

level (the horizontal black dot line in Fig. 5) as the area of sea ice leads is integrated for one month. The first significant correlation occurs when extending the integration time period to mid-to-late February (at day 49, r = -0.67, >99% significance). After that, the magnitude of the correlation gradually increases and the strongest relationship is achieved as the integration extended to early April (r=-0.73 at day 100). Extending the integration time period beyond early April does not improve the correlation. The evolution of the correlation coefficient between time series of sea ice leads and sea ice extent in

August (green line in Fig. 5) and September (red line in Fig. 5) is similar to that of July sea ice extent, but the relationship is not statistically significant at 95% confidence level, i.e., the largest correlation are -0.41 and -0.30 in early April for August and September, respectively.

       Next, we study the potential of the area of sea ice leads integrated from mid-winter to early spring can be used as a predictor of July, August and September sea ice extent, respectively. First, a linear regression model is used to calculate to

the dependent variable (the de-trended Arctic sea ice extent, ASIE) using the independent variable (the de-trended area of sea ice leads integrated from 1 January to 30 April, SILA), the linear regression is written as:

$$ASIE_{month} = A + B * SILA + e$$

where A and B denote the intercept and the slope of the least squares regression line and e is the residual or error.

       Figure 6a shows the regressed July Arctic sea ice extent anomalies. It appears that the observed interannual

variability of July ASIE anomalies can be reasonably reproduced by the area of sea ice leads that is integrated from January to April. The regression error (root mean square error, RMSE) decreases gradually as the integration time period increases (blue line in Fig. 6d). The smallest error (0.28 million $km^2$) occurs in April 10, which is smaller than the standard deviation of the observed July sea ice extent during 2003-2015 (0.54 million $km^2$). Figure 6b and 6c show the regressed August and September sea ice extent anomalies. The regressed August sea ice extent anomalies are off by a large margin for many years

during 2003-2015 as compared to the observations. This is also true for the regressed September sea ice extent anomalies. By the end of April, the error is 0.44 and 0.57 million $km^2$ for August (green line in Fig. 6d) and September (red line in Fig. 6d) respectively, which are comparable to the standard deviation of the observed ones (0.60 and 0.73 million $km^2$ for August and September).

The above regression analysis is applied to all the data during 2003-2015 to obtain the slope and intercept of the linear regression model. Next, we conduct the prediction using the linear regression model. Specifically, only the data from the first six years (2003-2008) are utilized to determine the slope and intercept of the linear regression model, and then Arctic sea ice extent anomalies during 2009-2015 are predicted using the corresponding integrated area of sea ice leads from January to April as inputs for the linear regression model. For July sea ice extent prediction (Fig. 6e), the evolution of the predicted ASIE anomalies is similar to the result of the aforementioned regression (the observed variability of July ASIE anomalies during 2009-2015 are well captured). As shown in Figure 6h (blue line), the prediction error decreases gradually as the integration time period increases, and the error reaches 0.28 million km$^2$ by the end of April which is smaller than the standard deviation of the observed July sea ice extent anomalies. For August and September sea ice extent prediction, the predicted sea ice extent anomalies cannot capture the observed ASIE anomalies, and the error is 0.51 and 0.63 million km$^2$ by the end of April for August and September, respectively. We also utilize the data from all previous years to determine the slope and intercept of the linear regression model, and then calculate the Arctic sea ice extent anomalies during 2009-2015, i.e., the predicted July sea ice extent anomalies in 2009 (2015) is based on the training using the data from 2003-2008 (2003-2014). The result of the predicted July sea ice extent anomalies is very similar to that using the data from the first six years (not shown).

Besides RMSE, the forecast skill (S) can be measured as follows:

$$S = 1 - \frac{\sigma_f}{\sigma_r}$$

where $\sigma_f$ is the RMSE of the prediction error and $\sigma_r$ is the RMSE of the observed July, August and September sea ice extent anomalies (with trend), respectively (0.54, 0.60 and 0.73 million km$^2$ during 2003-2015). S that is equal to 1 means a perfect prediction, equal to and less than 0 implies no prediction skill. As shown in Figure 6i, the prediction skill gradually increases with lengthening integration period. For July sea ice extent prediction, the predictive skill becomes the highest in late April (0.49). By contrast, there is no predictive skill for August and September sea ice extent. We also repeat this analysis by using the data from all previous years to determine the slope and intercept of the linear regression model. The result of the prediction skill is similar to Figure 6 (not shown).

In terms of the total Arctic sea ice extent, the integrated area of sea ice leads has a strong impact on July sea ice extent, but minor impacts on August and September sea ice extent. As shown in Figure 4 (4$^{th}$ row), the areas having significant negative correlations are mainly concentrated in the Atlantic and central and west Siberian sector of the Arctic. Here the Atlantic and central and west Siberian sector of the Arctic is defined from 15°W to 135°E (hereafter referred to as region-ATLCWS). We further examine the potential of the integrated area of sea ice leads in region-ATLCWS as a predictor of July, August and September sea ice extent in region-ATLCWS. We generate Figure 7 and 8 following the same procedures to generate Figure 5 and 6. For July sea ice extent in region-ATLCWS, the correlation increases as the integration time period increases (blue line in Fig. 7). The strongest relationship occurs at day 68 (r=-0.78, > 99%

significance) and then tends to level-off until the end of April. For August sea ice extent (green line in Fig. 7), the evolution of the correlation coefficient is similar to that of July sea ice extent and the correlation can reach to -0.57 (> 95% significance) and levels off until the end of April, which is much higher than that of pan-Arctic sea ice extent (r=-0.41). For September sea ice extent, though the relationship is better than the pan-Arctic result, the correlation is not statistically significant at 95% confidence level.

Following similar procedures in Figure 6, we calculate the regression and prediction analyses, except that the area of sea ice leads in region-ATLCWS is integrated from January 1 to April 30. The results show that the observed interannual variability of July and August sea ice extent anomalies in region-ATLCWS can be reasonably reproduced (Fig. 8a and 8b). The RMSE decreases gradually as the integration time period increases (blue and green lines in Fig. 8d). The smallest error occurs at day 68 for July (0.15 million km$^2$) and day 68 (0.13 million km$^2$) for August, which is smaller than the standard deviation of the observed sea ice extent during 2003-2015 (0.33 million km$^2$ for July and 0.23 million km$^2$ for August). The observed September sea ice extent anomalies in region-ATLCWS cannot be reproduced using the integrated sea ice leads in the same region. In terms of the prediction, as shown in Figure 8h, the prediction error decreases gradually as the integration time period increases, and the error is 0.14 million km$^2$ for July and 0.10 million km$^2$ for August by the end of April. For September sea ice extent prediction, the predicted sea ice extent anomalies cannot capture the observed anomalies.

**4 Conclusion**

The Arctic sea ice extent through the melt season is known to strongly depend on the state of sea ice in winter and spring. In this study, we explore the potential of the integrated area of sea ice leads in the Arctic Ocean as a predictor for Arctic sea ice extent during the melt season. We find that the area of pan-Arctic sea ice leads integrated from mid-winter to late spring has a significant impact on the evolution of the pan-Arctic sea ice state midway through the melting season, having the potential to improve the prediction of July pan-Arctic sea ice extent. However, they cannot be used to improve predictive skill for August and September pan-Arctic sea ice extent. When the area of sea ice leads integrated in the Atlantic and central and west Siberian sector of the Arctic is used, the result shows good predictive skills for both July and August sea ice extent in the Atlantic and central and west Siberian sector of the Arctic.

To further ensure that the significant relationship between the area of sea ice leads and July sea ice extent is related to the area of actually present sea ice leads, rather than 1) cloud cover and 2) open water/polynyas in the marginal ice zone can be wrongly classified as sea ice leads. First, we examine the relationship between the area of clouds in the Arctic Ocean from late winter to mid-spring and Arctic sea ice extent during the melting season. Following the same procedure applied to the calculation of sea ice leads as shown above, the area of clouds is defined as the sum of the product of the cloud fraction and the area of the grid box (625 km$^2$) using the MOD08 daily cloud fraction data projected on the NSIDC polar-stereographic grid (25 km). We then calculate correlation coefficients between the de-trended time series of the integrated

the area of 1) clouds and 2) open water at each grid point and the de-trended time series of the total sea ice extent in July, respectively. Figure 9a shows significant correlations that exceed the 95% confidence level for clouds. It appears that the region having significant correlations associated with the cloud area is very different from that of sea ice leads, and the overlapped significant correlations only occurs in a small area as shown by grey crosses. We further calculate the correlation

between time series of the area of clouds integrated to the day given and time series of July Arctic sea ice extent. Note that time series of the area of clouds or area of open water is calculated over the region where sea ice leads and extent have significant correlations except the overlapped area (orange color in Fig. 9a). As shown in Figure 9b, there is no significant correlation between the cloud area and July sea ice extent throughout the entire period. Second, the area of open water is defined as the sum of the product of the open water fraction and the area of the grid box ($625 \ km^2$). We repeat the above

analysis. Again, only scattered areas have significant correlations associated with the open water area, and the overlapped significant correlation only occurs in a small area (Fig. 10a). There is no significant correlation between the open water area and July sea ice extent (Fig. 10b). This suggests that the significant relationship between the area of sea ice leads and July sea ice extent is related to the area of actually present sea ice leads, rather than cloud cover or marginal ice zone open water.

        Although the potential of sea ice leads in the prediction of basin wide and regional sea ice extent, sea ice leads are

largely unrepresented process in numerical prediction systems and climate models due to their highly nonlinear, small-scale, and intermittent features. As a result, potential effects of sea ice leads on Arctic prediction are not well understood. Given that sea ice leads strongly influence energy budget at atmosphere-sea ice-ocean interface and the statistical results from this study, it would stand to reason that understanding the role of sea ice leads in Arctic prediction can identify performance limitations of numerical prediction systems and climate models and yield routes for significant improvements.

*Acknowledgements.* This research is supported by the National Key R&D Program of China (2018YFA0605901 and 2016YFC1402704), the NOAA Climate Program Office (NA14OAR4310216), the NSFC (41176169), the China Scholarship Council and the Fundamental Research Funds for the Central Universities (312231103). We also thank the NSIDC and PANGAEA for providing the data used in this study.

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

**Figure Captions**

*Figure 1.* (a) MODIS cloud fraction (%), (b) SAR backscatter coefficient image, and (c) MODIS sea ice leads in the highlighted area (the northern Beaufort Sea) as shown by the box in (d) on April 11, 2015.

*Figure 2.* Time series of the daily area of pan-Arctic sea ice leads from January 1 to April 30 for the period of 2003-2015. The black solid line is the averaged area during 2003-2015 and the grey shaded denotes one standard deviation of interannual variability.

*Figure 3.* (a) Time series of the area of pan-Arctic sea ice leads averaged from January to April for the period of 2003-2015 and Arctic sea ice extent of July(blue), August(green) and September(red). (b) Spatial distribution of the trend of the sea ice leads area ($km^2$ per year). The black cross denotes statistically significant trend (> 95% confidence level).

*Figure 4.* Spatial distribution of significant negative correlations between the area of sea ice leads integrated for 30 days (1st row), 60 days (2nd row), 90 days (3rd row) and 120 days (4th row), and July (1st column), August (2nd column) and September (3rd column) sea ice extent, respectively. The areas with correlation exceeding 95% confidence level are marked with black crosses and the color shaded is the averaged area of sea ice leads for the given period.

*Figure 5.* Evolution of correlation coefficients between the total area of sea ice leads integrated from January 1 to April 30 and the total Arctic sea ice extent in July (blue line), August (green line) and September (red line) during 2003-2015. The horizontal black dot line is 99% confidence level.

*Figure 6.* Regressed the total Arctic sea ice extent anomalies (million $km^2$) in (a) July, (b) August and (c) September on the area of sea ice leads integrated from January 1 to April 30 and (d) the evolution of their regression errors; Predicted the total Arctic sea ice extent anomalies (million $km^2$) in (e) July, (f) August and (g) September based on the area of sea ice leads integrated from January 1 to April 30, (h) the evolution of their prediction errors and (i) their forecast skills. The blue, green and red lines are July, August and September, respectively.

*Figure 7.* Evolution of correlation coefficients between the region-ATLCWS area of sea ice leads integrated from January 1 to April 30 and the region-ATLCWS Arctic sea ice extent in July (blue line), August (green line) and September (red line) during 2003-2015. The horizontal black dot line is 99% confidence level.

*Figure 8.* Regressed the region-ATLCWS Arctic sea ice extent anomalies (million $km^2$) in (a) July, (b) August and (c) September on the region-ATLCWS area of sea ice leads integrated from January 1 to April 30 and (d) the evolution of their regression errors; Predicted the region-ATLCWS Arctic sea ice extent anomalies (million $km^2$) in (e) July, (f) August and (g) September based on the region-ATLCWS area of sea ice leads integrated from January 1 to April 30, (h) the evolution of their prediction errors and (i) their forecast skills. The blue, green and red lines are July, August and September, respectively.

*Figure 9.* (a) Spatial distribution of significant correlations between the area of clouds (blue) and sea ice leads (orange) integrated from 1 January to 30 April with July sea ice extent. Grey cross denotes the overlapped significant correlations. (b) Evolution of correlation coefficients between the total area of cloud integrated from January 1 to April 30 and the total Arctic sea ice extent in July (blue line) during 2003-2015. The horizontal line is 99% (black dot) confidence level.

*Figure 10.* (a) Spatial distribution of significant correlations between the area of open water (blue) and sea ice leads (orange) integrated from 1 January to 30 April with July sea ice extent. Grey cross denotes the overlapped significant correlations; (b) Evolution of correlation coefficients between the total area of open water integrated from January 1 to April 30 and the total Arctic sea ice extent in July (blue line) during 2003-2015. The horizontal line is 99% (black dot) confidence level.

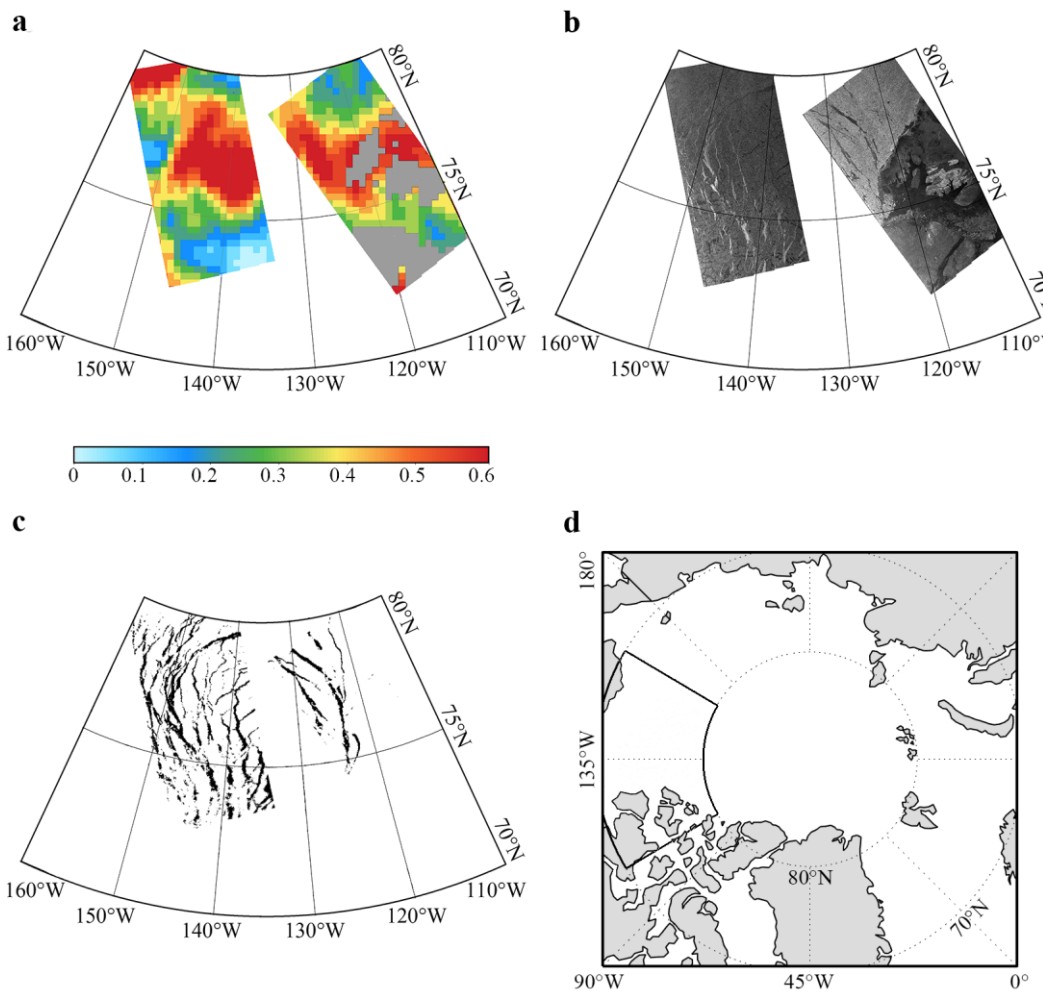

**Figure 1. (a) MODIS cloud fraction (%), (b) SAR backscatter coefficient image, and (c) MODIS sea ice leads in the highlighted area (the northern Beaufort Sea) as shown by the box in (d) on April 11, 2015.**

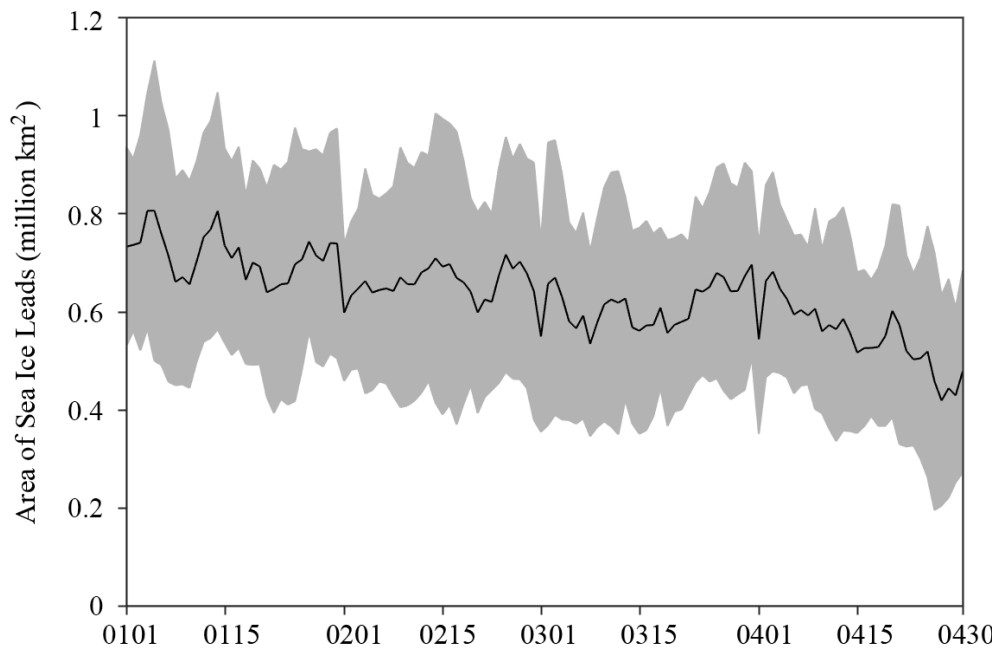

**Figure 2. Time series of the daily area of pan-Arctic sea ice leads from January 1 to April 30 for the period of 2003-2015. The black solid line is the averaged area during 2003-2015 and the grey shaded denotes one standard deviation of interannual variability.**

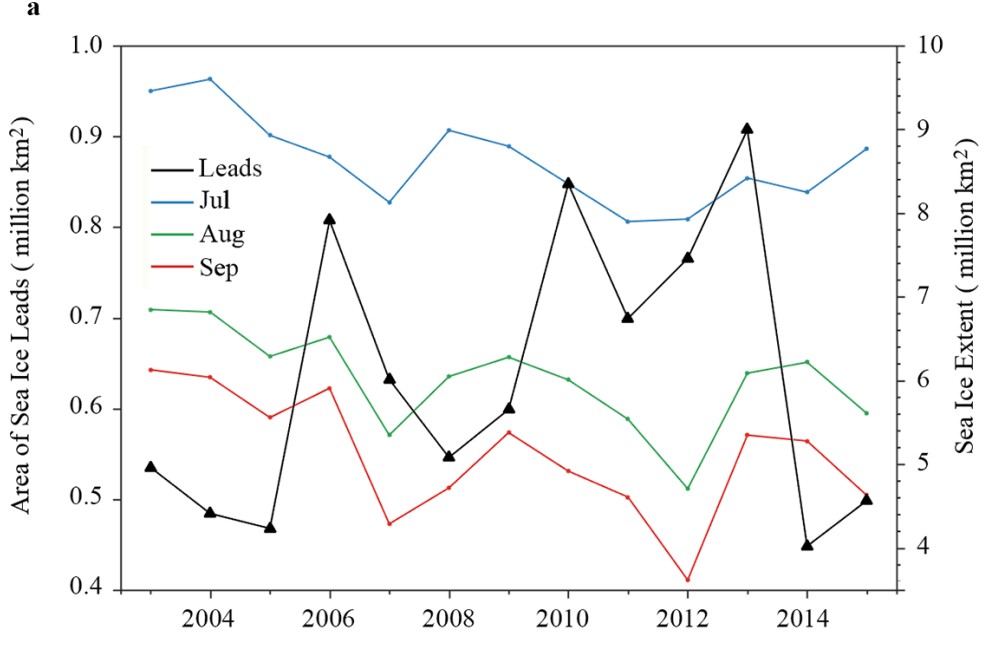

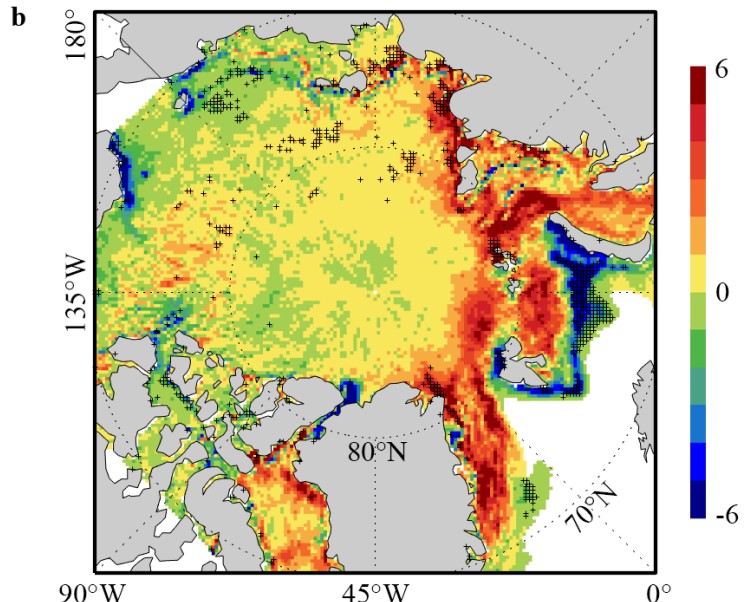

**Figure 3. (a) Time series of the area of pan-Arctic sea ice leads averaged from January to April for the period of 2003-2015 and Arctic sea ice extent of July(blue), August(green) and September(red). (b) Spatial distribution of the trend of the sea ice leads area (km² per year). The black cross denotes statistically significant trend (> 95% confidence level).**

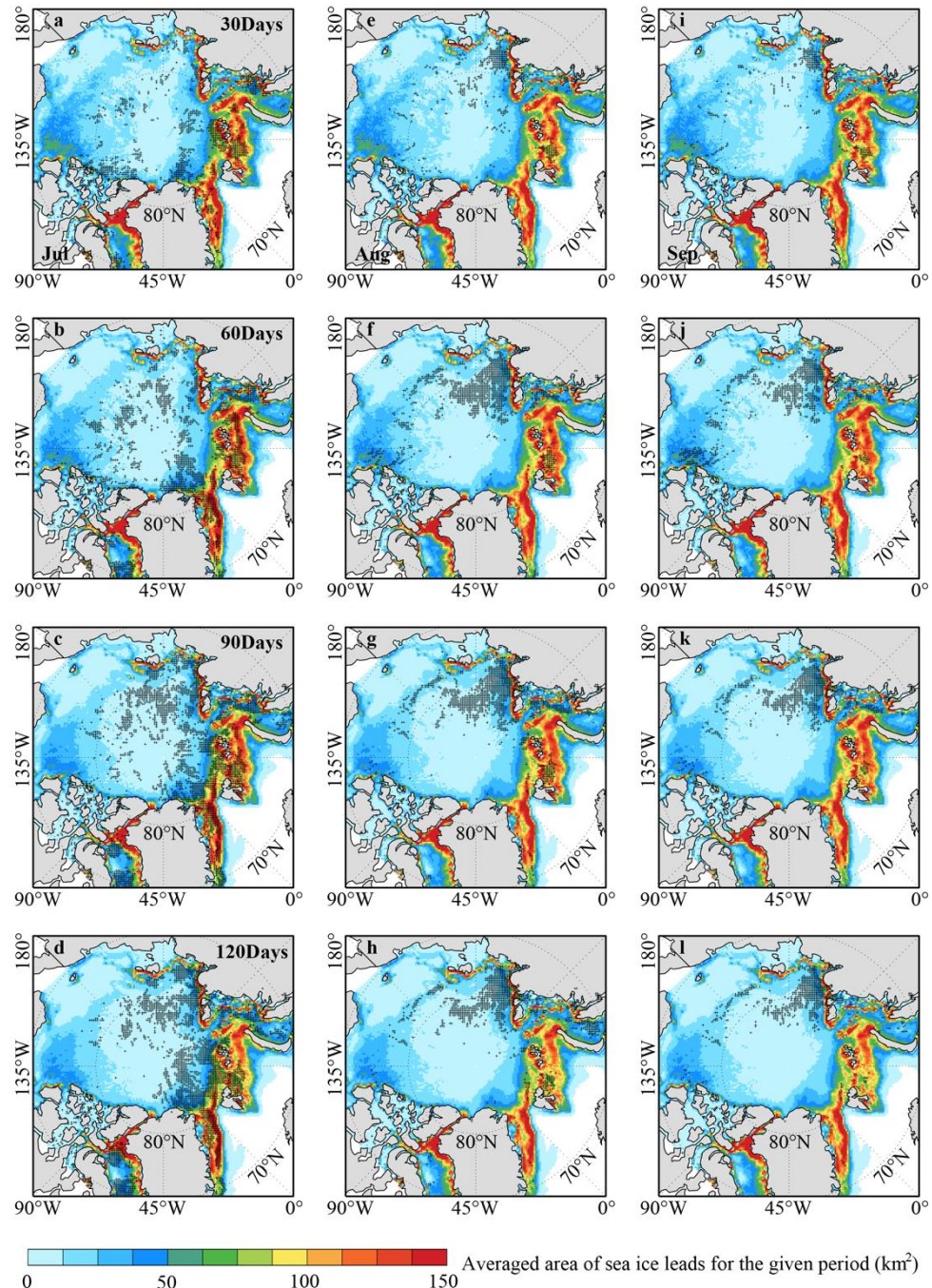

**Figure 4. Spatial distribution of significant negative correlations between the area of sea ice leads integrated for 30 days (1ˢᵗ row), 60 days (2ⁿᵈ row), 90 days (3ʳᵈ row) and 120 days (4ᵗʰ row), and July (1ˢᵗ column), August (2ⁿᵈ column) and September (3ʳᵈ column) sea ice extent, respectively. The areas with correlation exceeding 95% confidence level are marked with black crosses and the color shaded is the averaged area of sea ice leads for the given period.**

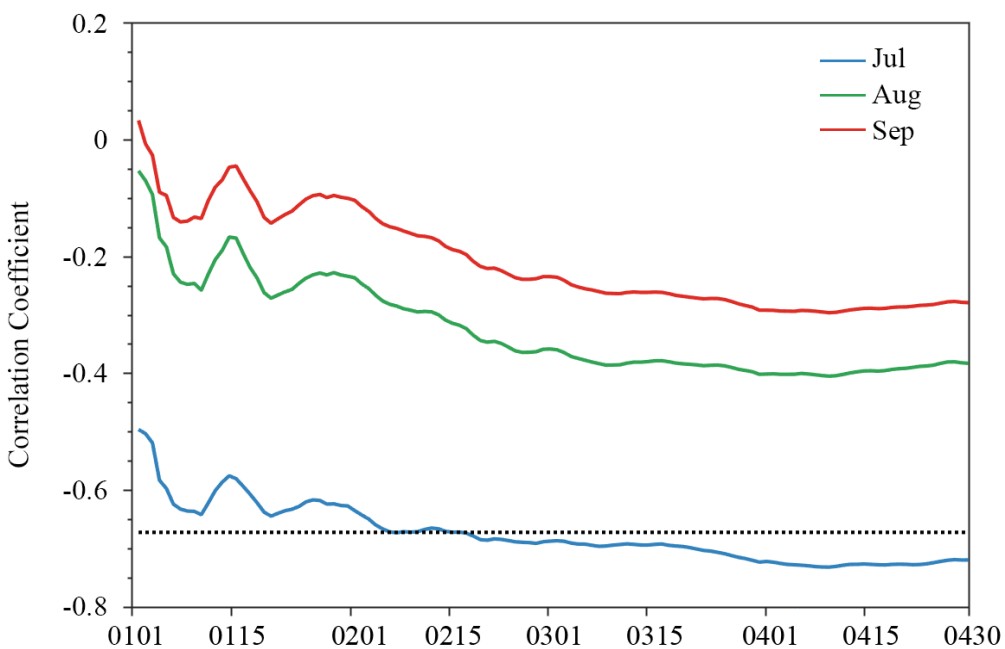

**Figure 5. Evolution of correlation coefficients between the total area of sea ice leads integrated from January 1 to April 30 and the total Arctic sea ice extent in July (blue line), August (green line) and September (red line) during 2003-2015. The horizontal black dot line is 99% confidence level.**

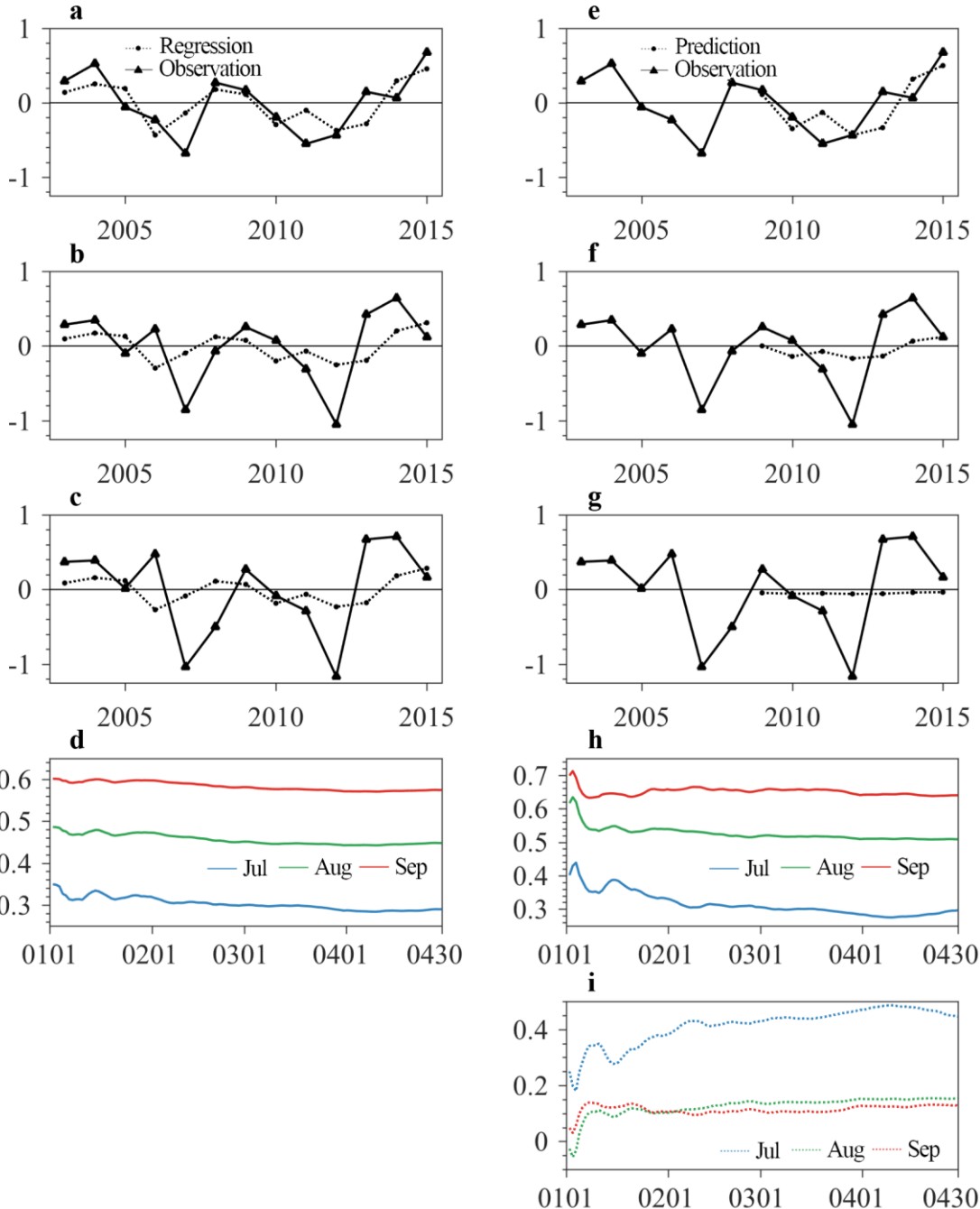

**Figure 6. Regressed the total Arctic sea ice extent anomalies (million km²) in (a) July, (b) August and (c) September on the area of sea ice leads integrated from January 1 to April 30 and (d) the evolution of their regression errors; Predicted the total Arctic sea ice extent anomalies (million km²) in (e) July, (f) August and (g) September based on the area of sea ice leads integrated from January 1 to April 30, (h) the evolution of their prediction errors and (i) their forecast skills. The blue, green and red lines are July, August and September, respectively.**

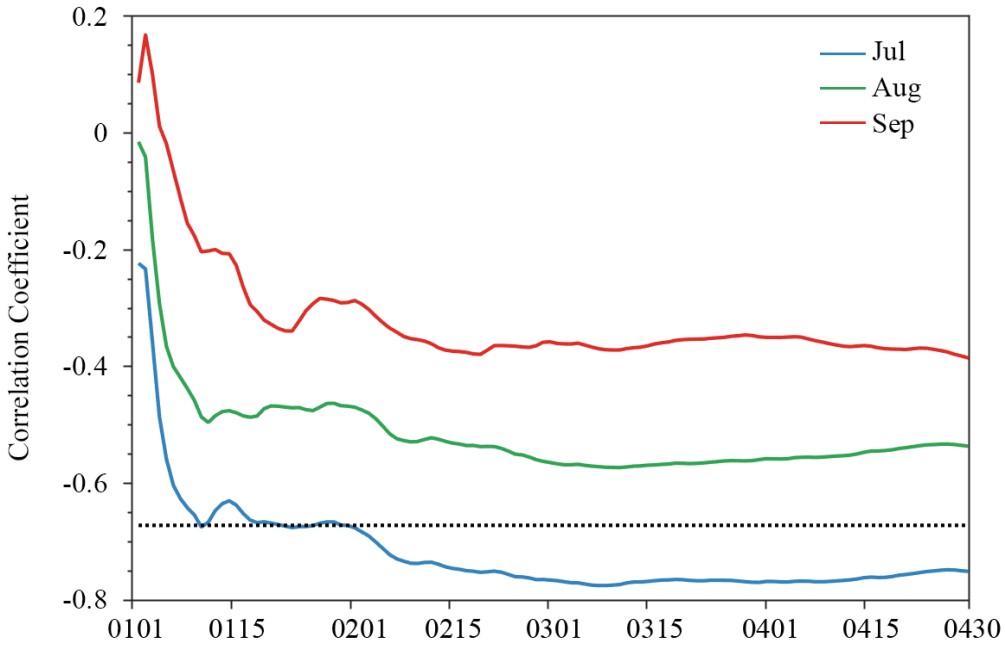

**Figure 7. Evolution of correlation coefficients between the region-ATLCWS area of sea ice leads integrated from January 1 to April 30 and the region-ATLCWS Arctic sea ice extent in July (blue line), August (green line) and September (red line) during 2003-2015. The horizontal black dot line is 99% confidence level.**

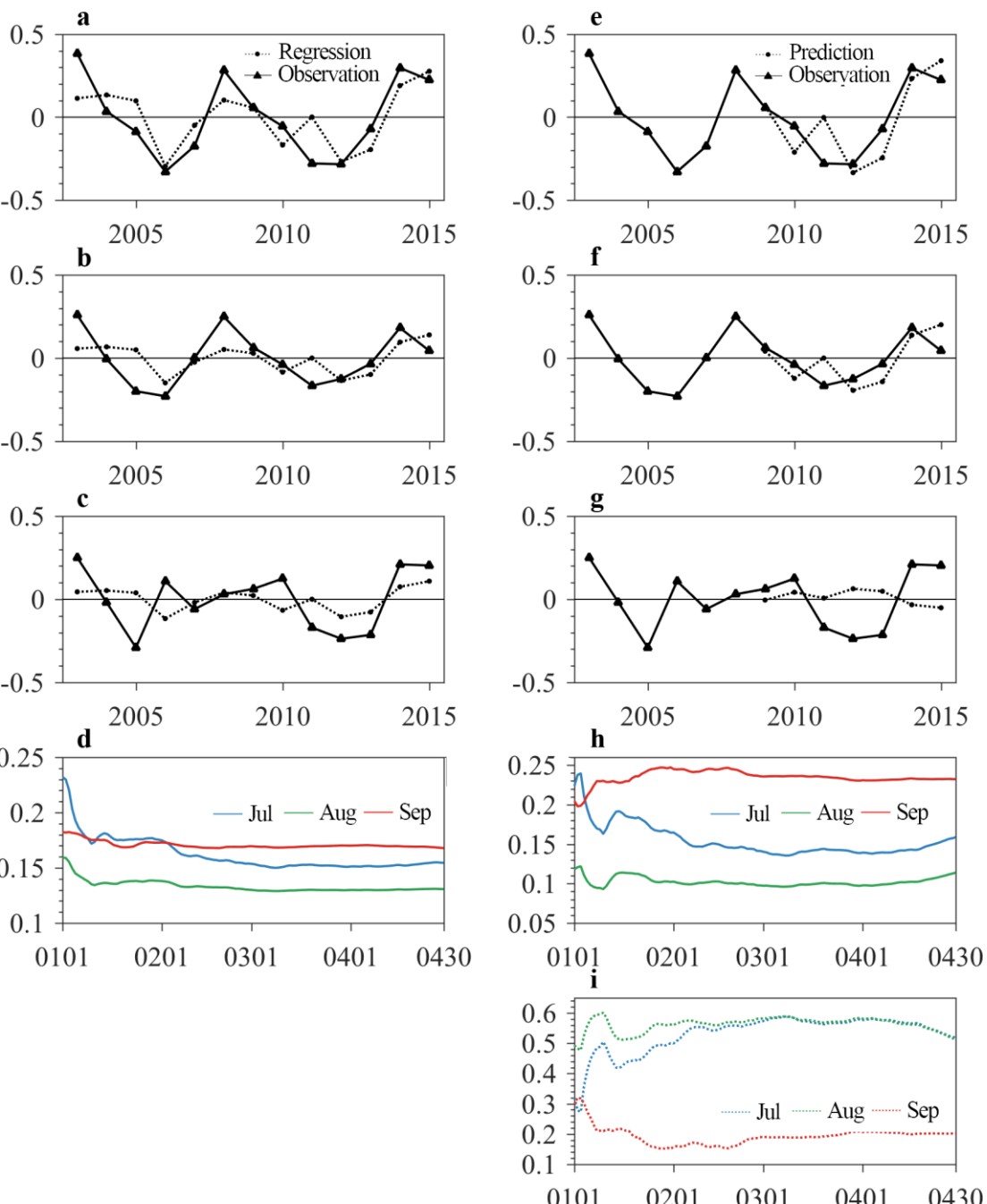

**Figure 8. Regressed the region-ATLCWS Arctic sea ice extent anomalies (million km$^2$) in (a) July, (b) August and (c) September on the region-ATLCWS area of sea ice leads integrated from January 1 to April 30 and (d) the evolution of their regression errors; Predicted the region-ATLCWS Arctic sea ice extent anomalies (million km$^2$) in (e) July, (f) August and (g) September based on the region-ATLCWS area of sea ice leads integrated from January 1 to April 30, (h) the evolution of their prediction errors and (i) their forecast skills. The blue, green and red lines are July, August and September, respectively.**

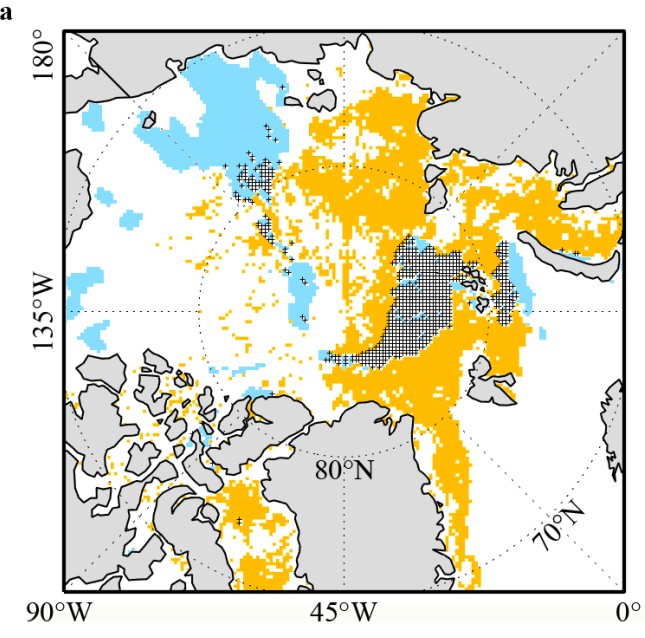

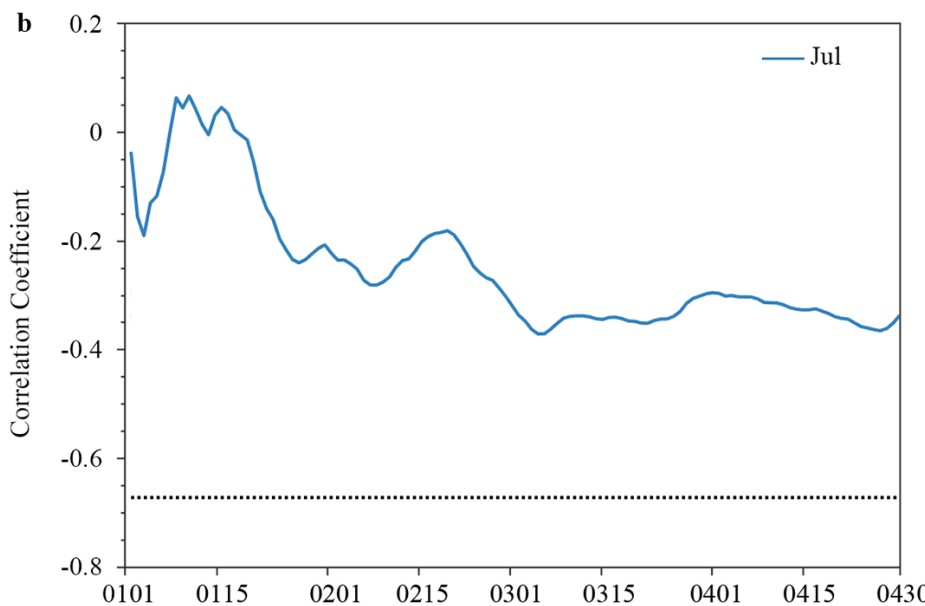

**Figure 9. (a) Spatial distribution of significant correlations between the area of clouds (blue) and sea ice leads (orange) integrated from 1 January to 30 April with July sea ice extent. Grey cross denotes the overlapped significant correlations. (b) Evolution of correlation coefficients between the total area of cloud integrated from January 1 to April 30 and the total Arctic sea ice extent in July (blue line) during 2003-2015. The horizontal line is 99% (black dot) confidence level.**

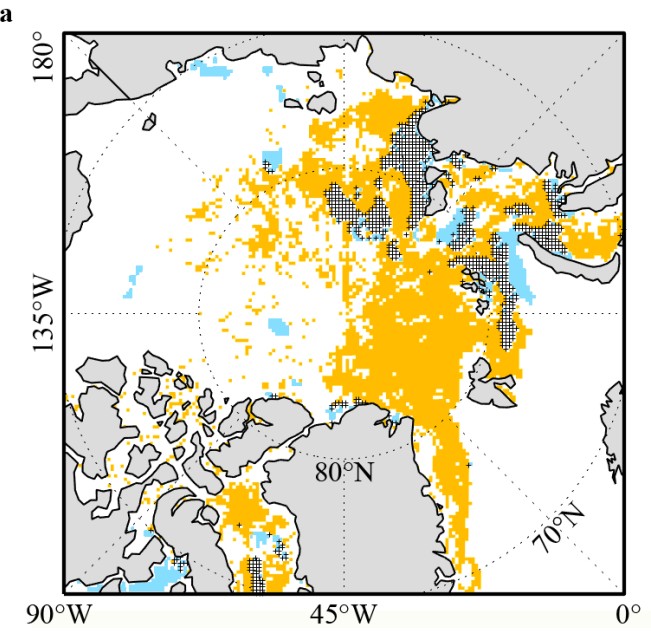

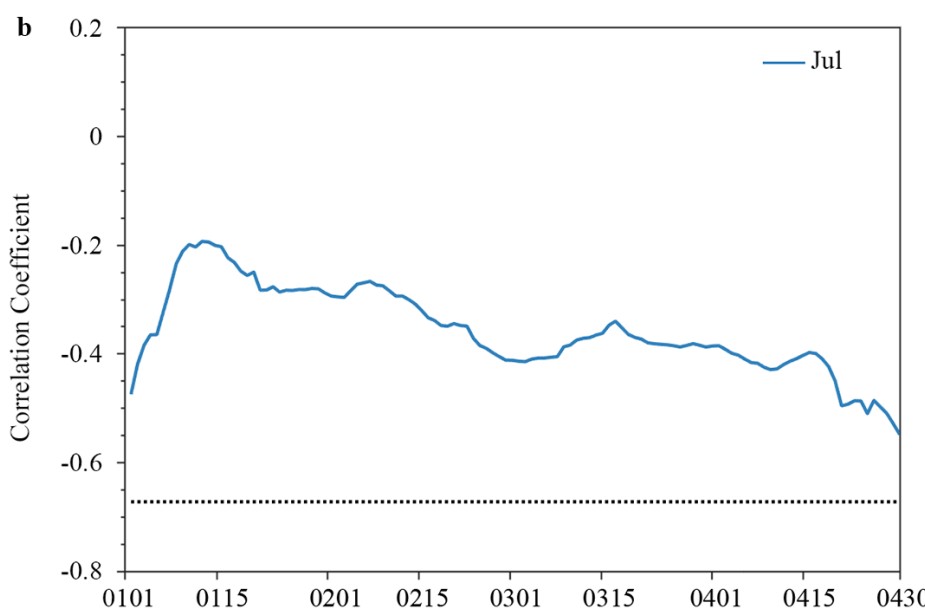

**Figure 10. (a) Spatial distribution of significant correlations between the area of open water (blue) and sea ice leads (orange) integrated from 1 January to 30 April with July sea ice extent. Grey cross denotes the overlapped significant correlations; (b) Evolution of correlation coefficients between the total area of open water integrated from January 1 to April 30 and the total Arctic sea ice extent in July (blue line) during 2003-2015. The horizontal line is 99% (black dot) confidence level.**