# Peer review of "The potential of sea ice leads as a predictor for summer Arctic sea ice extent"

_The Cryosphere, 2018_

## Referee Comment (RC1) · Anonymous Referee #1 · 17 Aug 2018

The manuscript investigates a linear regression between the area of Arctic sea-ice leads in January to April and pan-Arctic and regional monthly-mean sea-ice extent in July to September. The sea-ice lead area (SILA) is derived by the authors from MODIS satellite infrared observations and covers the years 2003-2015, and the sea-ice extent (SIE) is provided by satellite passive microwave. The authors find that the January-to-April SILA is significantly correlated with the July pan-Arctic SIE, but not with the August or September SIE. If SILA and SIE are restricted to the sector 15W - 135E, correlations of SILA and SIE are significant for both July and August. The authors claim that these correlations can be exploited to accurately predict summer SIE from late-winter SILA, which they present as the main conclusion of the manuscript.

The manuscript makes some interesting points about the statistical relation between

the area of Arctic sea-ice leads (SILA) observed in winter and the Arctic sea-ice extent observed in the following summer. As a side effect, it also makes statements about interannual variability and trends of SILA that are worth recording. The results are well presented, with clearly structured and text and high-quality figures. I particularly enjoyed reading the introduction which gives a good overview over related literature. However, there are some doubts about the validity of the main conclusions drawn in the manuscript, because (1) there clearly are ambiguities with the calculation of SILA from the infra-red observational data set that are not mentioned at all in the manuscript, (2) only very vague explanations of physical mechanisms are offered that can plausibly explain the correlation between SILA and SIE, and (3) the prediction results (right column of Figure 5) to me seem overly confident. I will detail my concerns (1) - (3) in the general comments below. I would recommend publication of this interesting manuscript after a substantial revision that fully addresses these doubts and convinces me of the validity of the main conclusions.

General comments:

1a) MODIS infrared observations of the surface are only availabe under cloud-free conditions. Therefore, it is potentially misleading to directly calculate the pan-Arctic or regional area of sea-ice leads from the gridded observational product as done by the authors. A brief look at one season of daily gridded maps of the sea-ice lead data product reveals that a large fraction of the sea-ice covered area is obscured by clouds for almost every day, and as expected there are large day-to-day variations in the cloud cover. Therefore, it is not clear at all how SILA as calculated by the authors relates to the area of actually present sea-ice leads. What if the year-to-year variability of SILA shown in Figure 2a is actually dominated by the variability in cloud cover obscuring a constant actual lead area to varying degrees? Varying cloud cover would be an alternative explanation for varying summer ice extent, because winter-time clouds keep the surface warm and inhibit sea-ice growth. The role of clouds needs to be properly discussed before a robust conclusion about the lead area can be drawn.
1b) It is also evident from the gridded maps of the sea-ice lead product that polynyas and the marginal ice zone in the Atlantic sector are wrongly classified as leads. It might well be that year-to-year variability in the area of polynyas and the width of the marginal ice zone in the Atlantic sector is responsible for the year-to-year variability in the SILA calculated by the authors. This would then invalidate their main conclusion as it is specific to sea-ice leads. Please provide some further analysis that quantifies how much of the SILA signal comes from polynyas and the Atlantic marginal ice zone.

2) Related to point (1) above, it would make the author's main conclusions more credible if they were supported by independent observational data, modelling results, or process studies. I would leave it up to the authors to decide what is most appropriate. An idea would be to have a look at observational products of cloud cover on the one hand, and an observational product of winds and sea-ice drift on the other hand. The first is important for thermodynamic ice growth, the second for the creation of leads. From studying the inter-annual variability of clouds, sea-ice drift and winds, some support or additional doubt could be derived regarding the author's main conclusions.

3) I am a bit sceptical about the skill achieved in "prediction mode" as shown in the right-hand column of Figure 5. For example, the forecasts shown in Figure 5e are almost identical to the regressed values shown in Figure 5a. This is surprising given the moderate amount of correlation in the time series used to construct the linear predictor. Could it be that the authors accidentally used the complete time series to construct the linear predictor, rather than only the first 6 years? Can the authors please check their analysis and provide further evidence that the prediction results in Figures 5e-g have been calculated exactly as described in the text?

4) The Data Section needs a more detailed description of the MODIS sea-ice leads data set. This description needs to also discuss the limitations and assumptions of the data set. This comment is related to points 1a) and 1b) above. Furthermore, I would suggest to rename the section to "Data and Methods" and move lines 9-16 of page 4 to that section. The description of how the SILA is calculated needs to include more

details on how clouds and artifacts in the observational data set are treated.

Specific comments:

1) In the title, the last word "prediction" is a duplication of "predictor" and needs to be removed. "seasonal" should be replaced by "summer", because only the months July-September are considered. 2) In the abstract, line 14, the wording "accurately predicted" is subjective and ambiguous. Please provide numbers. 3) The quantity defined on page 7 is not a forecast skill, but rather a potential forecast skill. A forecast skill (score) is always based on comparing the skill of the forecast with the skill of a reference forecast (e.g. climatology, or a linear trend forecast). I would suggest that in this case comparison with a linear-trend would be appropriate, e.g. S = 1 - RMSE(SILA regression) / RMSE(trend). See Jollife and Stephenson (2012) for an introduction into forecast verification.

References:

Jolliffe, I. T., & Stephenson, D. B. (Eds.). (2012). Forecast Verification: A Practicioner's Guide in Atmospheric Science. Wiley-Blackwell.
* * *

---

## Referee Comment (RC2) · Anonymous Referee #2 · 24 Aug 2018

**Review on "The potential of sea ice leads as a predictor for seasonal Arctic sea ice extent prediction" by Zhang et al.**

Recommendation: Publish after revisions

This paper reports some interesting work that will be a valuable contribution to the literature of the study of Arctic sea ice. The authors try to use sea ice leads as a predictor for the future sea ice extent and show that July pan-Arctic sea ice extent can be accurately predicted from the area of sea ice leads integrated from mid-winter to late spring. I would recommend it to be published with revisions subject to the comments below:

1. There are many places where the English could be improved (also refer to the specific comments below).
2. The key finding of this work is "to use sea ice leads as a predictor for the future sea ice extent", it would be useful if the authors can demonstrate that by using 2003-2015 data, they predict sea ice 2013-2015 sea ice extent and verify it. This would be more convincing.
3. Based on Figure 2, it seems that there is very large interannual variability of sea ice leads for January-April (for example, there is about 50% reduction from 2013 to 2014), it would be useful to add the sea ice extent for July, August, and September for 2003-2015 in Figure 2.

Specific comments:

- P1 Line 29: "north hemisphere" - change it to "northern hemisphere".
- P2 Line 10: "parameters can significantly improve" - change it to "parameters can significantly contribute to the improvement in".
- P2 Line 25: "depend strongly" - change it to "depends strongly".
- P2 Lines 27-28: "In additional, the albedo of sea ice leads" - change it to "In addition, the albedo of leads"
- P6 Line 10: "black solid line" - there seems no black solid line in Fig. 4. Either change the word here or add the black solid line in Fig. 4.
- P14 Line 5: "2013-2015" – change it to "2003-2015".

---

## Author Comment (AC1) · 1 Oct 2018

**Response to the reviews of TC-2018-108 "The potential of sea ice leads as a predictor for seasonal Arctic sea ice extent prediction" by Yuanyuan Zhang, Xiao Cheng, Jiping Liu, and Fengming Hui**

We greatly appreciate the thoughtful comments from the reviewers. According to the reviewer's comments, we revised the original manuscript.

**Responses to reviewer #1 comments**

Thank you very much for your careful reviewing of our manuscript. All issues raised have been considered thoroughly. The point-to-point response to the issues is appended below.

*General comments:*

*Question 1a) MODIS infrared observations of the surface are only available under cloud-free conditions. Therefore, it is potentially misleading to directly calculate the pan-Arctic or regional area of sea-ice leads from the gridded observational product as done by the authors. A brief look at one season of daily gridded maps of the sea-ice lead data product reveals that a large fraction of the sea-ice covered area is obscured by clouds for almost every day, and as expected there are large day-to-day variations in the cloud cover. Therefore, it is not clear at all how SILA as calculated by the authors relates to the area of actually present sea-ice leads. What if the year-to-year variability of SILA shown in Figure 2a is actually dominated by the variability in cloud cover obscuring a constant actual lead area to varying degrees? Varying cloud cover would be an alternative explanation for varying summer ice extent, because winter-time clouds keep the surface warm and inhibit sea-ice growth. The role of clouds needs to be properly discussed before a robust conclusion about the lead area can be drawn.*

*Question 1b) It is also evident from the gridded maps of the sea-ice lead product that polynyas and the marginal ice zone in the Atlantic sector are wrongly classified as leads. It might well be that year-to-year variability in the area of polynyas and the width of the marginal ice zone in the Atlantic sector is responsible for the year-to-year variability in the SILA calculated by the authors. This would then invalidate their main conclusion as it is specific to sea-ice leads. Please provide some further analysis that quantifies how much of the SILA signal comes from polynyas and the Atlantic marginal ice zone.*

*Question 2) Related to point (1) above, it would make the author's main conclusions more credible if they were supported by independent observational data, modelling results, or process studies. I would leave it up to the authors to decide what is most appropriate. An idea would be to have a look at observational products of cloud cover on the one hand, and an observational product of winds and sea-ice drift on the other hand. The first is important for thermodynamic ice growth, the second for the creation of leads. From studying the inter-annual variability of clouds, sea-ice drift and winds, some sup- port or additional doubt could be derived regarding the author's main conclusions.*

**Response:**

We appreciate reviewer's suggestions. Since question 1a, 1b and 2 are related, here we address them together.

1) We agree with the reviewer that cloud contamination is a major issue plaguing the retrieval of the pan-Arctic sea ice leads from the MODIS infrared observation. To address this issue, Willmes and Heinemann (2015a) used multi-temporal satellite images that make full use of cloud-free pixels and assumed that changes of surface characteristics are insignificantly over synoptic time scale for detecting sea ice leads in pixels obscured by clouds. The probability of a clear-sky view within a day is increased associated with the convergence of satellite tracks in high latitudes. As shown in Willmes and Heinemann (2015b), on average, the Arctic has a clear-sky frequency of 30-60% in the daily aggregates. The lowest availability of clear-sky data is in the Chukchi Sea (see Figure 3 in Willmes and Heinemann, 2015b for details). To further mitigate the issue of cloud contamination, William and Heinemann (2015a, b) implemented a fuzzy cloud artifact filter that employs temporal and spatial object characteristics to distinguish between physical sea ice leads and artifacts that arise from clouds.

References:
Willmes, S. and Heinemann, G.: Pan-Arctic lead detection from MODIS thermal infrared imagery, Annals of Glaciology, 56, 29-37, 2015a.
Willmes, S. and Heinemann, G.: Sea-ice wintertime lead frequencies and regional characteristics in the Arctic, 2003–2015, Remote Sensing, 8, 4, 2015b.

2) To further address the reviewer's concern, we compared the MODIS sea ice leads data used in this study with the Synthetic Aperture Radar (SAR) images under cloudy conditions. Compared to MODIS that receives thermal emissions or reflected components, SAR allows for penetration through most clouds and precipitation. We calculated backscatter coefficients from the Sentinel-1A Extra-Wide swath HH polarization images using the Sentinel Application Platform and projected them on the NSIDC polar-stereographic grid with a spatial resolution of 100 m. Cloudy conditions are determined using the MOD08 Level3 daily cloud fraction product (Hubanks et al., 2018). For example, Figure 1 and 2 show the MODIS cloud fraction, SAR backscatter coefficient image, and MODIS sea ice leads in the northern Beaufort Sea on April 11, 2015 and the central Arctic Ocean that is northeast of Greenland on April 9, 2015, respectively. Compared to SAR images, the MODIS sea ice leads data can capture the correct spatial distribution of sea ice leads under cloudy conditions. The consistence between the MODIS sea ice leads data and SAR images gives us more confidence about this data. Thus, the sea ice lead area (SILA) calculated in this study is related to the area of actually present sea ice leads.

Reference:
Hubanks, P., Platnick, S., King, M., and Ridgway, B.: MODIS atmosphere L3 gridded product algorithm theoretical basis document ATBD & Users Guide Reference Number: ATBD-MOD-30, Collection 006, Version 4.3, 128, 2018.

[Figure]

Figure 1. (a) MODIS cloud fraction (%), (b) SAR backscatter coefficient image, and (c) MODIS sea ice leads in the highlighted area (the northern Beaufort Sea) as shown by the box in (d) on April 11, 2015.

[Figure]

Figure 2. (a) MODIS cloud fraction (%), (b) SAR backscatter coefficient image, and (c) MODIS sea ice leads in the highlighted area (the central Arctic Ocean that is northeast of Greenland) as shown by the box in (d) on April 9, 2015.

3) More importantly, we examined the relationship between the area of clouds in the Arctic Ocean from late winter to mid-spring and Arctic sea ice extent during the melting season. Following the same procedure applied to the calculation of sea ice leads in the manuscript, the area of clouds is defined as the sum of the product of the cloud fraction and the area of the grid box (625 km$^2$) using the MOD08 daily cloud fraction data projected on the NSIDC polar-stereographic grid (25km). We then calculated correlation coefficients between the de-trended time series of the integrated the area of clouds at each grid point and the de-trended time series of the total sea ice extent in July. Figure 3 shows significant correlations that exceed the 95% confidence level. It appears that the region having significant correlations associated with the cloud area is very different from that of sea ice leads, and the overlapped significant correlations only occurs in a small area as shown by grey crosses in Figure 3. We further calculated the correlation between time series of the area of clouds integrated to the day given and time series of July Arctic sea ice extent. Note that time series of the area of clouds is calculated over the region where sea ice leads and extent have significant correlations

except the overlapped area (orange color in Figure 3). As shown in Figure 4, there is no significant correlation between the cloud area and July sea ice extent throughout the entire period. By contrast, significant correlation between the area of sea ice leads and July sea ice extent first occurs in mid-to-late February, the magnitude of the correlation gradually increases and the strongest relationship is achieved as the integration extended to early April (see Figure 4 in the original manuscript). This suggests that the significant relationship between the area of sea ice leads and July sea ice extent is related to the area of actually present sea ice leads, rather than cloud cover.

[Figure]

Figure 3. Spatial distribution of significant correlations between the area of clouds (blue) and sea ice leads (orange) integrated from 1 January to 30 April with July sea ice extent. Grey cross denotes the overlapped significant correlations.

[Figure]

Figure 4. Evolution of correlation coefficients between the total area of cloud integrated from January 1 to April 30 and the total Arctic sea ice extent in July (blue line) during 2003-2015. The horizontal line is 99% (black dot) confidence level.

4) As suggested by the reviewer, open water/polynyas in the marginal ice zone can be wrongly classified as sea ice leads. Although the retrieval method of Willmes and Heinemann (2015a) is based on significant positive surface temperature anomalies associated with the presence of a lead with respect to its surrounding area, they removed swath-level temperature gradients by deriving the local temperature anomalies based on the temperature distribution in 51 x 51 kernel. This tends to reduce the misclassification of leads and open water/polynyas. Their data is limited to January to April because their method relies on a pronounced thermal contrast between leads and open water/polynyas. For example, Figure 5 shows the MODIS sea ice leads fraction and open water fraction computed from the NASA Team sea ice concentration data (25 km) on April 30, 2015. To make the two data comparable, the 1.5 km MODIS sea ice leads data is projected on the NSIDC polar-stereographic grid with a spatial resolution of 25 km. Clearly, sea ice leads fraction in the marginal ice zone in the Chukchi and Beaufort Seas (Figure 5b) and the Greenland, Iceland and Norwegian (GIN) Seas (Figure 5d) is different from that of open water, (Figure 5a and c), and the magnitude of sea ice leads fraction is much smaller than that of open water.

[Figure]

Figure 5. Spatial distribution of open water fraction (left column) and sea ice leads fraction (right column) in the marginal ice zone in the Chukchi and Beaufort Seas and the GIN Seas on April 30, 2015.

5) To further address the reviewer's concern, we examined the relationship between the area of open water calculated from NASA Team sea ice concentration data in the Arctic Ocean from late winter to mid-spring and Arctic sea ice extent during the melting season. Following the same procedure applied to the calculation of sea ice leads in the manuscript and clouds above, the area of open water is defined as the sum of the product of the open water fraction and the area of the grid box ($625 \text{ km}^2$). We then calculated correlation coefficients between the de-trended time series of the integrated the area of open water at each grid points and the de-trended time series of the total sea ice extent in July. Figure 6 shows significant correlations that exceed the 95% confidence level. It appears that scattered areas have significant correlations associated with the open water area, and the overlapped significant correlation only occurs in a small area as grey crosses in Figure 6. We further calculated the correlation between time series of the area of open water integrated to the day given and time series of July Arctic sea ice extent. Note that time series of the area of open water is calculated over the regions where sea ice leads and sea ice extent have significant correlations except overlapped area (orange color in Figure 6). As shown in Figure 7, there is no significant correlation between the open water area and July sea ice extent. By contrast, significant correlation between the area of sea ice leads and July sea ice extent first occurs in mid-to-late February, the magnitude of the correlation gradually increases and the strongest relationship is achieved as the integration extended to early April (see Figure 4 in the

original manuscript). This suggests that the significant relationship between the area of sea ice leads and July sea ice extent is related to the area of actually present sea ice leads, rather than open water/ polynyas.

[Figure]

Figure 6. Spatial distribution of significant correlations between the area of open water (blue) and sea ice leads (orange) integrated from 1 January to 30 April with July sea ice extent. Grey cross denotes the overlapped significant correlations.

[Figure]

Figure 7. Evolution of correlation coefficients between the total area of open water integrated from January 1 to April 30 and the total Arctic sea ice extent in July (blue line) during 2003-2015. The horizontal line is 99% (black dot) confidence level.

***Question 3)*** *I am a bit sceptical about the skill achieved in "prediction mode" as shown in the right-hand column of Figure 5. For example, the forecasts shown in Figure 5e are almost identical to the regressed values shown in Figure 5a. This is surprising given the moderate amount of correlation in the time series used to construct the linear predictor. Could it be that the authors accidentally used the complete time series to construct the linear predictor, rather than only the first 6 years? Can the authors please check their analysis and provide further evidence that the prediction results in Figures 5e-g have been calculated exactly as described in the text?*

**Response:**

We double checked our calculation. For the prediction analysis in the original manuscript, we actually used the data from all previous years to determine the slope and intercept of the linear regression model, and then calculated the predicted Arctic sea ice extent anomalies for the years of 2009-2015, instead of only the data of the first six years (2003-2008). For example, the predicted July sea ice extent anomaly in 2009 (2015) is based on the training using the data of 2003-2008 (2003-2014). Here we recalculated the prediction analysis for the years of 2009-2015 by only using the data of the first six years (2003-2008) to determine the slope and intercept of the linear regression model. As shown in Figure 8, the result of the predicted July sea ice extent anomalies (Fig. 8f) is very similar to that in the original manuscript (Fig. 8a), and predictive skill is even slightly better (Fig. 8e and 8j). There is still no predictive skill for August and September sea ice extent.

[Figure]

Figure 8. Predicted total Arctic sea ice extent anomalies (million km$^2$) in (a,f) July, (b,g) August and (c,h) September during 2009-2015 based on the area of sea ice leads integrated from January 1 to April 30, (d,i) the evolution of their prediction errors and (e,j) their forecast skills; Left column: the data of all previous years are used; Right column: only the data from the first six years (2003-2008);The blue, green and red lines are July, August and September, respectively.

***Question 4)** The Data Section needs a more detailed description of the MODIS sea-ice leads data set. This description needs to also discuss the limitations and assumptions*

*of the data set. This comment is related to points 1a) and 1b) above. Furthermore, I would suggest to rename the section to "Data and Methods" and move lines 9-16 of page 4 to that section. The description of how the SILA is calculated needs to include more details on how clouds and artifacts in the observational data set are treated.*

**Response:**

Based on the reviewer's suggestion, we renamed the section to "Data and Methods", and provided more information about the MODIS sea ice leads data set. We also moved the description of how to calculate the area of sea ice leads (SILA) to this section. Now this section reads as "…In a recent study, Willmes and Heinemann (2015a) presented a non-parameterized global threshold method, which was validated and applied to derive sea ice leads maps from surface temperature anomalies in the Arctic Ocean using the MODIS ice surface temperature product. Daily sea ice leads composites were created. The composite maps indicate the presence of cloud artifacts in the leads identification that arise from ambiguities in the MODIS cloud mask. To mitigate these artifacts, they implemented a fuzzy filter system that employs spatial and temporal object characteristics to distinguish between physical leads and artifacts. This approach advances the potential to retrieve daily leads maps operationally from the MODIS infrared product.

In this study, the pan-Arctic sea ice leads data is obtained from the Data Publisher for Earth & Environment Science (PANGAEA), which is available for the months from January to April for the period 2003-2015 (Willmes and Heinemann, 2015b). The spatial resolution of the daily binary sea ice leads map is about 1.5 km with omission 5% that can reflect sea ice leads variability except the Chukchi Sea (Willmes and Heinemann, 2015a, c), because clear-sky day is less than 15% in the Chukchi Sea. Cloud contamination is a major issue plaguing the retrieval of the pan-Arctic sea ice leads from the MODIS infrared observation. Here we compare the above MODIS sea ice leads data with the Synthetic Aperture Radar (SAR) images under cloudy conditions. Compared to MODIS that receives thermal emissions or reflected components, SAR allows for penetration through most clouds and precipitation. We calculate backscatter coefficients from the Sentinel-1A Extra-Wide swath HH polarization images using the Sentinel Application Platform and project them on the NSIDC polar-stereographic grid with a spatial resolution of 100 m. Cloudy conditions are determined using the MOD08 Level3 daily cloud fraction product (Hubanks et al., 2018). For example, Figure 1 shows the MODIS cloud fraction, SAR backscatter coefficient image, and MODIS sea ice leads in the northern Beaufort Sea on April 11, 2015. Compared to SAR images, the MODIS sea ice leads data can capture the correct spatial distribution of sea ice leads under cloudy conditions. The consistence between the MODIS sea ice leads data and SAR image gives us more confidence about this data.

The Arctic sea ice extent is obtained from the National Snow and Ice Data Center (NSIDC), which is derived from the Nimbus-7 Scanning Multichannel Microwave Radiometer, DMSP Special Sensor Microwave/Imager, and Special Sensor

Microwave Imager and Sounder sensors using NASA Team algorithm(Cavalieri et al., 1996, updated yearly).

The daily total area of sea ice leads is computed from the daily binary sea ice leads map, which is projected on the NSIDC polar-stereographic grid with a spatial resolution of 25 km. During the projection, we calculate the number of pixels with detected sea ice leads in a 25km grid box. Sea ice leads fraction is then defined as the ratio between the number of pixels with detected sea ice leads and the total number of pixels in the 25km grid box. The total area of sea ice leads is the sum of the product of the sea ice leads fraction and the area of the grid box ($625 \text{ km}^2$). Here the daily total area of sea ice leads is only calculated when the NSIDC sea ice concentration in the grid box is larger than 15% (commonly used as the threshold to define sea ice edge). ".

References:
Hubanks, P., Platnick, S., King, M., and Ridgway, B.: MODIS atmosphere L3 gridded product algorithm theoretical basis document ATBD & Users Guide Reference Number: ATBD-MOD-30, Collection 006, Version 4.3, 128, 2018.
Willmes, S. and Heinemann, G.: Pan-Arctic lead detection from MODIS thermal infrared imagery, Annals of Glaciology, 56, 29-37, 2015a.
Willmes, S. and Heinemann, G.: Daily pan-Arctic sea-ice lead maps for 2003-2015, with links to maps in NetCDF format, in 30 Supplement to: Willmes, S; Heinemann, G(2105): Sea-Ice Wintertime Lead Frequencies and Regional Characteristics in the Arctic, 2003-2015, Remote Sensing, 8(1),4, doi:10.3390/rs8010004, edited, PANGAEA, 2015b.
Willmes, S. and Heinemann, G.: Sea-ice wintertime lead frequencies and regional characteristics in the Arctic, 2003–2015, Remote Sensing, 8, 4, 2015c.

***Specific comments***

*1) In the title, the last word "prediction" is a duplication of "predictor" and needs to be removed. "seasonal" should be replaced by "summer", because only the months July-September are considered.*

**Response:**
We changed the title to "The potential of sea ice leads as a predictor for summer Arctic sea ice extent" and this issue has been checked throughout the manuscript.

*2) In the abstract, line 14, the wording "accurately predicted" is subjective and ambiguous. Please provide numbers.*

**Response:**
The sentence is changed to "Our results show that July pan-Arctic sea ice extent can be predicted from the area of sea ice leads integrated from mid-winter to late spring with the prediction error of 0.28 million $\text{km}^2$ that is smaller than the standard deviation of the observed interannual variability.".

*3) The quantity defined on page 7 is not a forecast skill, but rather a potential forecast skill. A forecast skill (score) is always based on comparing the skill of the forecast with the skill of a reference forecast (e.g. climatology, or a linear trend forecast). I would suggest that in this case comparison with a linear-trend would be appropriate, e.g. S=1-RMSE(SILA regression)/RMSE(trend). See Jollife and Stephenson (2012) for an introduction into forecast verification.*

**Response:**

Based on the reviewer's suggestion, we recalculated the forecast skill using the following equation:

$$S = 1 - \frac{\sigma_f}{\sigma_r}$$

where $\sigma_f$ is the RMSE of the prediction error and $\sigma_r$ is the RMSE of the observed July, August and September sea ice extent anomalies (with trend), respectively(0.54, 0.60 and 0.73 million km$^2$ during 2003-2015), respectively. As shown in Figure 9b, the evolution of forecast skills based on the reviewer's suggestion (S=1-RMSE(SILA regression)/RMSE(trend)) is similar to that of Figure 5i in the original manuscript (that is for July sea ice extent prediction, the predictive skill gradually increases with lengthening integration period and becomes the highest in late April). Again, there is no predictive skill for August and September sea ice extent.

[Figure]

Figure 9. The evolution of forecast skills based on the integrated area of sea ice leads starting from January 1, (a) S=1-MSE(SILA regression)/MSE(detrend), (b) S=1-RMSE(SILA regression)/RMSE(trend). The blue, green and red dot lines are July, August and September, respectively.

---

## Author Comment (AC2) · 1 Oct 2018

**Response to the reviews of TC-2018-108 "The potential of sea ice leads as a predictor for seasonal Arctic sea ice extent prediction" by Yuanyuan Zhang, Xiao Cheng, Jiping Liu, and Fengming Hui**

We greatly appreciate the thoughtful comments from the reviewers. According to the reviewer's comments, we revised the original manuscript.

**Responses to reviewer #2 comments**
Thank you very much for your careful reviewing of our manuscript. All issues raised have been considered thoroughly and a major revision as suggested has been complete. The point-to-point response to the issues mentioned is appended below.

*Question 1) There are many places where the English could be improved (also refer to the specific comments below).*

**Response:**
We improved English of the revised manuscript.

*Question 2) The key finding of this work is "to use sea ice leads as a predictor for the future sea ice extent", it would be useful if the authors can demonstrate that by using 2003-2015 data, they predict sea ice 2013-2015 sea ice extent and verify it. This would be more convincing.*

**Response:**
Based on the reviewer's suggestion, we did the following three analyses. For the first column of Figure 1, all the data during 2003-2015 are used to train the linear regression model, and then the predicted Arctic sea ice extent (SIE) anomalies are calculated for 2009-2015 (2013-2015 is marked with red cross). It shows that the observed interannual variability of July sea ice extent anomalies can be reasonably reproduced by the area of sea ice leads, except that the predicted anomaly in 2013 deviates from the observations substantially. For the second column of Figure 1, only the data from the first six years (2003-2008) are used to train the linear regression model, and then the predicted SIE anomalies is calculated for 2009-2015. The result of the predicted July SIE anomalies is very similar to those of the 1st column. For the third column of Figure 1, the data from all previous years are used to train the linear regression model, and then predicted SIE anomalies are calculated for 2009-2015, i.e., the predicted July SIE anomalies in 2009 (2015) is based on the training using the data from 2003-2008 (2003-2014). Again, the result of the predicted July SIE anomalies resembles those of the 1st and 2nd column. As shown by the 4th row of Figure 1, the prediction error by the end of April for the three analyses is much smaller than the standard deviation of the observed July SIE anomalies. Thus, the area of an-Arctic sea ice leads integrated from mid-winter to late spring has the potential to improve the prediction of July pan-Arctic SIE.

However, all three analyses show that there is still no predictive skill for August and September sea ice extent.

[Figure]

Figure 1. Predicted the total Arctic sea ice extent anomalies (million km$^2$) in July (first row), August (second row), September (third row) during 2009-2015 based on the area of sea ice leads integrated from January 1 to April 30, 1$^{st}$ column: all the data during 2003-2015 are used, 2$^{nd}$ column: only the data from the first six years (2003-2008) are used, and 3$^{rd}$ column: the data of all previous years are used. The fourth row is the evolution of the prediction errors. The blue, green and red lines are July, August and September, respectively.

*Question 3) Based on Figure 2, it seems that there is very large interannual variability of sea ice leads for January-April (for example, there is about 50% reduction from 2013 to 2014), it would be useful to add the sea ice extent for July, August, and September for 2003-2015 in Figure 2.*

**Response:**
Based on the reviewer's suggestion, we added July, August and September sea ice extent for the period of 2003-2015 in Figure 2. We also calculated the correlation coefficients between July, August, September sea ice extent and the area of sea ice leads averaged from January to April during 2003-2015, which are -0.51, -0.30 and -0.23, respectively. It appears that July sea ice extent is more closely related to the area of sea ice leads than August and September.

[Figure]

Figure 2. Time series of the area of the pan-Arctic sea ice leads averaged from January to April for the period of 2003-2015 (black line), and July, August, September sea ice extent during 2003-2015. The blue, green and red lines are July, August and September, respectively.

***Specific comments:***
- *P1 Line 29: "north hemisphere" - change it to "northern hemisphere".*

**Response:**
This sentence is changed to "…increases the frequency of abnormal weather and climate in the mid-latitude of the northern hemisphere and influences the thermohaline circulation…".

- *P2 Line 10: "parameters can significantly improve" - change it to "parameters can significantly contribute to the improvement in".*

**Response:**
This sentence is changed to "The results show that some parameters can significantly contribute to the improvement in seasonal sea ice forecast at different lead times".

- *P2 Line 25: "depend strongly" - change it to "depends strongly".*

**Response:**
This sentence is changed to "Sensible heat flux over sea ice leads depends strongly on leads' width.".

- *P2 Lines 27-28: "In additional, the albedo of sea ice leads" - change it to "In addition, the albedo of leads"*

**Response:**

The sentence is changed to "the albedo of leads is about 0.07 under cloudy condition…".

- *P6 Line 10: "black solid line" - there seems no black solid line in Fig. 4. Either change the word here or add the black solid line in Fig. 4.*

**Response:**
We have changed it to "black dot line".

- *P14 Line 5: "2013-2015" – change it to "2003-2015".*

**Response:**
We have changed it and checked throughout the manuscript.